# Learnable Expansion of Graph Operators for Multi-Modal Feature Fusion

**Dexuan Ding**[1]     **Lei Wang**[*, 1, 2]     **Liyun Zhu**[1]     **Tom Gedeon**[3]     **Piotr Koniusz**[2, 1]
[1]Australian National University, [2]Data61/CSIRO, [3]Curtin University

## Abstract

In computer vision tasks, features often come from diverse representations, domains (*e.g.*, indoor and outdoor), and modalities (*e.g.*, text, images, and videos). Effectively fusing these features is essential for robust performance, especially with the availability of powerful pre-trained models like vision-language models. However, common fusion methods, such as concatenation, element-wise operations, and non-linear techniques, often fail to capture structural relationships, deep feature interactions, and suffer from inefficiency or misalignment of features across domains or modalities. In this paper, we shift from high-dimensional feature space to a lower-dimensional, interpretable graph space by constructing relationship graphs that encode feature relationships at different levels, *e.g.*, clip, frame, patch, token, *etc*. To capture deeper interactions, we expand graphs through iterative graph relationship updates and introduce a learnable graph fusion operator to integrate these expanded relationships for more effective fusion. Our approach is relationship-centric, operates in a homogeneous space, and is mathematically principled, resembling element-wise relationship score aggregation via multilinear polynomials. We demonstrate the effectiveness of our graph-based fusion method on video anomaly detection, showing strong performance across multi-representational, multi-modal, and multi-domain feature fusion tasks.

## 1 Introduction

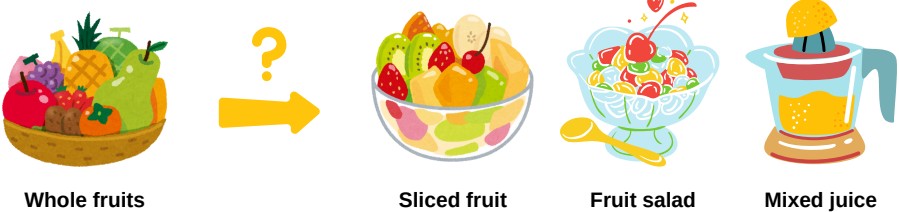

**Whole fruits**              **Sliced fruit**      **Fruit salad**      **Mixed juice**

Figure 1: **Can we squeeze more?** This figure shows feature fusion in computer vision, from whole fruits (raw features) to sliced fruit (early fusion) and fruit salad (late fusion). The juice represents our graph-based fusion approach, which mixes multi-modal data for richer insights.

Imagine preparing a fruit salad (see Figure 1). Initially, we slice fruits like apples, bananas, and oranges into distinct pieces, each retaining its unique flavor. This is analogous to features in multi-modal data, sourced from different modalities such as text, images, or videos. Combining these fruit slices resembles traditional early fusion methods in computer vision, where features are concatenated but remain largely independent of each other (Snoek et al., 2005; Gadzicki et al., 2020; Barnum et al., 2020). Next, we might cut the fruit into smaller pieces and mix them further, but the distinct flavors persist. This reflects late fusion methods, which combine outputs from separately trained models on different modalities (Snoek et al., 2005; Bodla et al., 2017; Wang et al., 2019a). While some integration occurs, the deeper interactions between the features are still missing, just as the flavors in the salad remain separate. Finally, we use a fruit mixer. This tool thoroughly blends

---

*Corresponding author (lei.w@anu.edu.au).

the fruits, creating a smooth, unified mixture where each flavor enhances the whole. This blending captures the essence of feature fusion. Our proposed graph-based fusion method parallels the fruit mixer, it doesn't just combine features but captures their complex, multi-level relationships. By focusing on interactions between feature relationships, we aim for a richer, more integrated fusion, revealing insights that traditional methods miss.

Traditional fusion techniques like concatenation, element-wise operations, or attention mechanisms (Dai et al., 2021) often capture shallow or superficial interactions. These approaches typically overlook deeper, structural relationships between feature elements (Atrey et al., 2010; Feng et al., 2019), limiting their ability to align features across different modalities or domains. Furthermore, they often suffer from inefficiencies in computation and alignment. Our motivation for this work arises from the limitations of current methods, which struggle to blend and enhance feature relationships meaningfully. We propose a paradigm shift from high-dimensional feature spaces to lower-dimensional, interpretable graph spaces. Instead of relying on raw features, our approach emphasizes the fusion of relationships between features, similar to how a fruit mixer blends distinct flavors into a cohesive whole. Specifically, we introduce relationship graphs, such as similarity graphs, as intermediary representations that encode the relationships between entities like frame-, patch-, or token-level features from videos. These graphs provide a more compact and interpretable representation of the data (Mai et al., 2020). In a similarity graph, nodes correspond to entities (*e.g.*, video clips, frames, or patches), while edges represent their relationships. To capture more complex interactions, we use iterative graph relationship updates to refine existing connections. This process reveals deeper structural insights that are often overlooked by traditional fusion methods.

We also introduce a learnable weight matrix, the graph fusion operator, which combines different graph relationship updates. Unlike simple concatenation or addition that treats all feature components equally, our learnable mechanism dynamically weights the contributions of different graph relationship updates, resulting in better fusion performance across various modalities, domains, and representations. Our graph-based fusion operates in a lower-dimensional, homogeneous space, offering several advantages. First, by representing relationships instead of individual features, it reduces dimensionality and computational costs. Second, the homogeneous space allows consistent fusion across domains and modalities, aligning features into a common structure. Third, it provides better interpretability by focusing on relationships rather than abstract features. The use of iterative graph relationship updates reveals refined feature interactions. Lastly, the learnable fusion mechanism adapts to specific tasks via learning objectives, improving both performance and efficiency.

Furthermore, our approach can be interpreted as a multilinear polynomial with learnable coefficients, where relationship scores are aggregated across iterative graph relationship update sequences. This mathematically grounded framework generalizes simple linear operations, capturing more complex interactions between graph relationship updates. Our main **contributions** are as follows:

i. We propose a novel graph-based fusion framework, termed EGO fusion, which effectively captures multi-representational, multi-modal, and multi-domain relationships through relationship graphs, thereby enriching feature representations.

ii. We introduce a learnable graph fusion operator that dynamically integrates different graph relationship updates, facilitating deeper interactions among features and balancing self-relationships with inter-feature relationships.

iii. We establish a theoretical connection between our graph fusion approach and multilinear polynomials, providing insights into feature interactions. We empirically validate our method in video anomaly detection, demonstrating improvements in both performance and interpretability over traditional feature-level fusion techniques.

## 2 RELATED WORK

**Traditional feature fusion.** Traditional fusion methods in multi-modal learning (D'mello & Kory, 2015) typically rely on simple operations such as concatenation, element-wise addition, or multiplication (Chen et al., 2023a). Early Fusion techniques (Snoek et al., 2005), for example, combine features from different modalities, such as text, images, and videos, into a single high-dimensional feature vector. While straightforward, this approach often leads to overfitting and increased computational complexity due to the high dimensionality of the concatenated features. Moreover, early

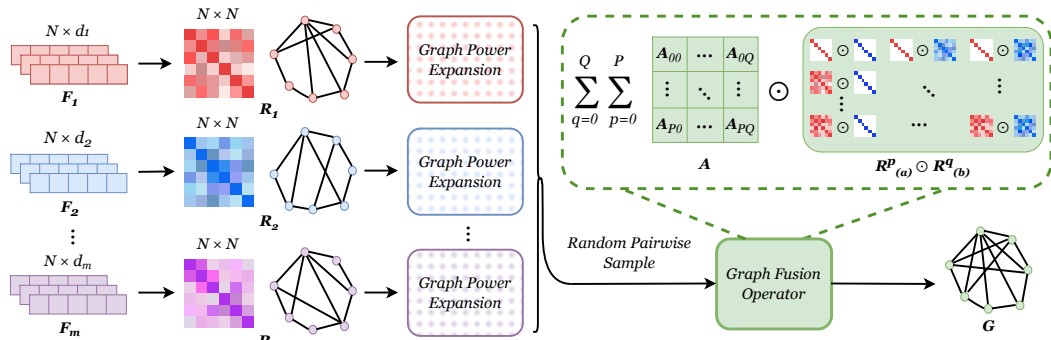

Figure 2: EGO fusion, our graph-based fusion framework, comprises three key components: (i) relationship graph reconstruction, (ii) graph expansion via element-wise multiplication, and (iii) a graph fusion operator (learnable $A$) that aggregates representations within a unified graph space.

fusion tends to amplify noise from heterogeneous data sources, negatively impacting performance in complex tasks (Liu et al., 2016). Late Fusion, on the other hand, merges the outputs of independently trained models from different modalities (Snoek et al., 2005). This approach alleviates some dimensionality issues, allowing each model to focus on learning modality-specific features before integration. However, it fails to capture deep interactions between modalities, limiting its effectiveness. More advanced techniques, such as attention mechanisms (Vaswani, 2017; Dai et al., 2021), have been introduced to dynamically weigh features based on importance. Nonetheless, these methods often overlook structural relationships between features, restricting their ability to model complex interactions. Neural network-based non-linear fusion introduces learnable layers to the process (Wang et al., 2019a; Wang & Koniusz, 2021), but these models often lack transparency, making it difficult to interpret how individual features contribute to predictions. Aligning features from disparate modalities remains a challenge, further complicating fusion of non-comparable data.

**Graph-based fusion.** To overcome the limitations of traditional methods, recent work has turned to graph-based approaches (Liao et al., 2013; Feng et al., 2019; Iyer et al., 2020; Chen & Zhang, 2020; Mai et al., 2020; Zhang et al., 2024), which emphasize the relationships between features rather than the features themselves. In these methods, data is represented as a graph, with nodes corresponding to entities such as frames, patches, or tokens, and edges capturing similarities or interactions (Iyer et al., 2020). This structured representation allows for more interpretable and context-aware feature fusion. Graph Convolutional Networks (GCNs) have been widely used for modeling relationships across modalities (Zhang et al., 2020), particularly in video understanding tasks (Huang et al., 2020; Gkalelis et al., 2021), where capturing temporal and spatial relationships is critical. GCNs aggregate information from neighboring nodes, enabling the modeling of context-dependent interactions. Despite their effectiveness, most graph-based methods capture only first-order relationships, limiting their ability to model complex, multi-step dependencies between features (Chen & Zhang, 2020).

Our work uses iterative graph relationship updates, which enable flexible and localized information fusion while preserving the graph structure. This enables our method to capture refined and more nuanced structural interactions, revealing insights that conventional fusion techniques often miss.

**Interpretable and efficient fusion.** As machine learning models grow in complexity, there is increasing demand for fusion methods that balance interpretability and computational efficiency. Traditional approaches like concatenation and neural-based fusion often operate in high-dimensional spaces, which can lead to inefficiencies and hinder transparency. Recent advancements have focused on designing more interpretable fusion strategies (Ma et al., 2016). For example, Capsule Networks (Sabour et al., 2017) and attention mechanisms (Vaswani, 2017) aim to provide greater insight into the fusion process by highlighting important features. However, these methods still suffer from the computational burdens associated with high-dimensional data, particularly in large-scale, multi-modal tasks.

Our approach offers a different solution by shifting from feature-level fusion to relationship-centric fusion, operating in a lower-dimensional graph space. This transition improves both interpretability

and efficiency, as it focuses on capturing feature relationships rather than raw features. Using iterative graph relationship updates, our method models deeper interactions among features, enabling more effective fusion while avoiding the high-dimensional computations of traditional methods. Additionally, our learnable graph fusion operator dynamically weights feature interactions, leading to a more adaptive and task-specific fusion process.

## 3  APPROACH

This section introduces our proposed method, Expansion of Graph Operators (EGO) fusion. We begin by defining key notations, followed by the construction of our relationship graph, the formulation of iterative graph relationship updates as graph expansions, and our fusion strategy.

**Notations.** Let $\mathcal{I}_T = 1, 2, \ldots, T$ represent the index set. Scalars are denoted by regular fonts, *e.g.*, $x$; vectors by lowercase boldface, *e.g.*, $\boldsymbol{x}$; matrices by uppercase boldface, *e.g.*, $\boldsymbol{X}$; and tensors by calligraphic letters, *e.g.*, $\mathbf{X}$.

### 3.1  EGO: EXPANSION OF GRAPH OPERATORS

> **Relationship graph of unit-level features.** Text, images, and videos can be used to extract various *unit-level* features (see definition in Appendix A), ranging from word- and paragraph-level to patch-, clip-, frame-, cube-, or token-level, using pre-trained models. These heterogeneous features are then transformed into a homogeneous graph space by modeling pairwise relationships among unit-level features, such as similarities, distances, or other relevant metrics. Since distances and similarities are inversely related, meaning high similarity corresponds to low feature distance (see proof in Appendix B), similarity scores are particularly effective for encoding local relationships among units, helping to identify which feature points are close or similar. In the resulting relationship graph, *e.g.*, similarity graph, each unit feature point is represented as a node, and the graph structure captures the local neighborhood of these unit-level features.

To show this process, we consider extracting unit-level feature representations from multiple pre-trained models or multi-modal sources. We denote these feature representations as $\boldsymbol{F}_{(1)} \in \mathbb{R}^{N \times d_1}$, $\boldsymbol{F}_{(2)} \in \mathbb{R}^{N \times d_2}$, ..., $\boldsymbol{F}_{(T)} \in \mathbb{R}^{N \times d_T}$. Here, $\boldsymbol{F}_{(m)} \in \mathbb{R}^{N \times d_m}$ ($m \in \mathcal{I}_T$) denotes the feature set from the $m$-th model or modality, where $N$ corresponds to the number of unit-level features, and each feature exists in $d_m$ dimensions. We begin by computing pairwise relationships between the unit-level features within each model or modality as follows:

$$s_{i,j} = r(\boldsymbol{f}_i, \boldsymbol{f}_j) \tag{1}$$

where $r(\cdot, \cdot)$ is a relationship function, *e.g.*, cosine similarity, Gaussian kernel, or another distance metric. The term $s_{i,j}$ represents the relationship score between unit-level features $\boldsymbol{f}_i$ and $\boldsymbol{f}_j$ from the feature set $\boldsymbol{F}$ (the model or modality index is omitted for simplicity). Using these pairwise relationships, we build a relationship graph represented by the matrix:

$$\boldsymbol{R} = [s_{i,j}]_{(i,j) \in \mathcal{I}_N \times \mathcal{I}_N} \tag{2}$$

where $\boldsymbol{R} \in \mathbb{R}^{N \times N}$ captures the pairwise relationships between unit-level features. A value of 1 in the matrix indicates a strong relationship, *e.g.*, two unit-level features are identical in visual or textural concepts, while a value of 0 means no relationship. This matrix serves as the adjacency matrix for the graph.

However, while similarity- or distance-based graphs are useful, they often fail to capture the global structure of the data, as they predominantly rely on local information. To address this limitation, we propose a novel graph expansion approach based on iterative graph relationship updates. This method enhances the graph's representation by expanding graph relationships in a more controlled and localized manner. Through iterative refinement, nodes can incorporate richer information.

**Graph powers *vs*. Iterative graph relationship updates.** Graph powers refer to the repeated multiplication of a graph's adjacency matrix, which shows multi-step connections between nodes. For a graph represented by the adjacency matrix $\boldsymbol{R}$, the $k$-th power of the graph, denoted as $\boldsymbol{R}^k$, uncovers relationships between nodes that are $k$ steps apart. Specifically, each element $\boldsymbol{R}_{i,j}^k$ indicates

the cumulative influence of all paths of length $k$ between nodes $i$ and $j$. This mechanism is useful for modeling long-range dependencies in a graph. However, our approach focuses on iterative graph relationship updates using element-wise multiplication rather than traditional graph powers with matrix multiplication. Instead of matrix exponentiation, we iteratively refine the relationship graph through a series of adaptive updates, providing more direct control over feature propagation. This process enables the model to dynamically capture both local and global dependencies without the constraints imposed by power-based adjacency transformations.

To operationalize this, consider two distinct relationship graphs $\boldsymbol{R}_{(a)}$ and $\boldsymbol{R}_{(b)}$. We construct a sequence of iterative relationship updates for each model or modality:

$$
\begin{cases}
\mathbf{G}_{(a)} = \left[ \boldsymbol{R}_{(a)}^0, \boldsymbol{R}_{(a)}^1, \cdots, \boldsymbol{R}_{(a)}^P \right] \in \mathbb{R}^{N \times N \times (P+1)} \\
\mathbf{G}_{(b)} = \left[ \boldsymbol{R}_{(b)}^0, \boldsymbol{R}_{(b)}^1, \cdots, \boldsymbol{R}_{(b)}^Q \right] \in \mathbb{R}^{N \times N \times (Q+1)}
\end{cases}, \tag{3}
$$

where $\boldsymbol{R}_{(a)}^p$ and $\boldsymbol{R}_{(b)}^q$ represent the relationship graphs obtained after $p$ and $q$ iterative updates for $\boldsymbol{R}_{(a)}$ and $\boldsymbol{R}_{(b)}$ respectively. The initial relationship graph $\boldsymbol{R}^0$ serves as a baseline, typically preserving self-connections via an identity matrix, *i.e.*, $\boldsymbol{R}^0 = \boldsymbol{I}$.

These sequences reflect how information propagates between nodes over multiple refinement steps, offering insights into both local and global relationships within the graph. Unlike static graph power expansions, our iterative approach dynamically adjusts node interactions at each step, allowing for a more flexible and expressive feature integration process.

**Graph fusion operator.** We introduce a novel graph fusion operator, denoted as $\circledast$, designed to integrate information from different modalities by merging their relationship graphs through iterative updates. This operator enhances the model's capacity to learn complex representations by incorporating both direct and higher-order relationships within the data. Mathematically, the graph fusion is expressed as follows:

$$
\boldsymbol{G} = \mathbf{G}_{(a)} \circledast \boldsymbol{A} \circledast \mathbf{G}_{(b)}^\mathsf{T}
$$
$$
= \sum_{q=0}^{Q} \sum_{p=0}^{P} \boldsymbol{R}_{(a)}^p a_p \odot \boldsymbol{R}_{(b)}^q b_q
$$
$$
= \sum_{q=0}^{Q} \sum_{p=0}^{P} a_p b_q \left( \boldsymbol{R}_{(a)}^p \odot \boldsymbol{R}_{(b)}^q \right), \tag{4}
$$

where $\boldsymbol{a} = [a_p]_{p \in \mathcal{I}_{(P+1)}}$ and $\boldsymbol{b} = [b_q]_{q \in \mathcal{I}_{(Q+1)}}$ are the modality graph update selectors, and $\boldsymbol{A} = \boldsymbol{a} \otimes \boldsymbol{b} \in \mathbb{R}^{(P+1) \times (Q+1)}$, with $\otimes$ representing the outer product. We also propose an advanced variant:

$$
\boldsymbol{G} = \sum_{q=0}^{Q} \sum_{p=0}^{P} \boldsymbol{A}_{p,q} \left( \boldsymbol{R}_{(a)}^p \odot \boldsymbol{R}_{(b)}^q \right), \tag{5}
$$

where $\boldsymbol{A} \in \mathbb{R}^{(P+1) \times (Q+1)}$ is a learnable weight matrix that modulates the fusion process. The operator $\odot$ denotes element-wise multiplication. In Appendix C, we provide a detailed explanation of our graph fusion process, including the derivation of equation 5, and the relationship between equation 4 and equation 5. The fused relationship graph $\boldsymbol{G} \in \mathbb{R}^{N \times N}$ is the result of this integration, enabling the model to optimize the combination of graph relationship updates through backpropagation, thus improving the fusion of features across different levels.

We observe that $\boldsymbol{R}_{(a)}^p \odot \boldsymbol{R}_{(b)}^q$ in equation 4 and equation 5 captures all possible combinations of iterative graph relationship updates, while $\boldsymbol{A}$ provides the appropriate weights for the fusion process. Incorporating the initial relationship graph, *e.g.*, $\boldsymbol{R}^0$, in the fusion process is critical; it preserves self-connections and maintains original feature information. This mechanism enables adaptive weighting, balancing the influence of direct relationships and progressively refined interactions, ensuring the model emphasizes the most relevant connections tailored to the specific task.

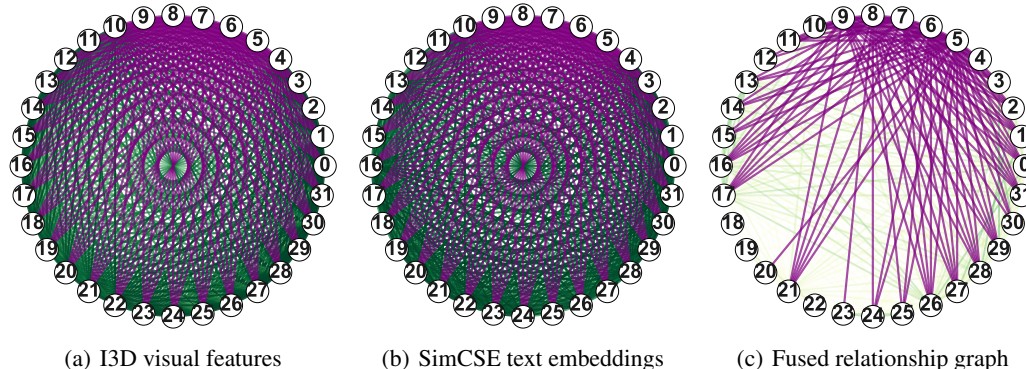

(a) I3D visual features      (b) SimCSE text embeddings      (c) Fused relationship graph

Figure 3: Comparison of relationship graphs on ShanghaiTech. The graphs are constructed using cosine similarity to represent relationships among features: (a) visual features, (b) text embeddings, and (c) the fused graph that integrates both modalities. In each graph, nodes represent clip-level (or unit-level; see Appendix A) features, with numbers indicating the sequence order of the video clips. Edges, shown in green, represent cosine similarity between features, with darker shades indicating stronger connections. Anomaly nodes and their connections are highlighted in purple (*e.g.*, the connection from node 4 to 10). The fused relationship graph, generated using our EGO fusion method, effectively integrates visual and textual information into a unified structure, resulting in fewer connections among abnormal nodes. This effect is achieved through our regularization term in equation 7, which encourages anomaly nodes to have fewer connections than normal nodes, regardless of connection strength. Appendix L includes additional visualizations.

To enhance the fusion of diverse features across varying representations, domains, and modalities, we implement a random sampling strategy during each training iteration. By randomly sampling two relationship graphs for the fusion process, we ensure that the integration occurs within a homogeneous graph space while still facilitating the amalgamation of features from disparate domains. This random sampling introduces variability and robustness into the fusion process, significantly boosting the model's capacity to learn meaningful representations. Furthermore, this strategy enriches the feature set and cultivates a comprehensive understanding of inter-relationships across different data modalities, ultimately leading to improved performance. Figure 3 compares the original relationship graphs for visual and textual features with our EGO-fused relationship graph.

## 3.2 CONNECTING TO MULTILINEAR POLYNOMIALS

Our graph fusion process is intrinsically connected to multilinear polynomials, which provide a robust mathematical framework for aggregating multiple powers of relationship scores between two relationship graphs. Specifically, each entry in the fused graph $G$ is a function of the element-wise relationship scores between the two input graphs. These scores are then combined in a manner that parallels how terms in a multilinear polynomial are constructed. This allows us to capture both linear and nonlinear interactions between modalities.

We can express each entry in the fused relationship graph, $G_{i,j}$, as a multilinear combination of the relationship scores from the two graphs, $G_{(a)}$ and $G_{(b)}$, at different powers. Rewriting equation 4 or equation 5 based on equation 1, we get:

$$G_{i,j} = A_{0,0} + A_{0,1}s_{(b)i,j} + A_{0,2}s^2_{(b)i,j} + \cdots + A_{P,Q}s^P_{(a)i,j}s^Q_{(b)i,j}$$
$$= \sum_{p=0}^{P}\sum_{q=0}^{Q} A_{p,q}s^p_{(a)i,j}s^q_{(b)i,j} \tag{6}$$

The term $A_{p,q}$ serves as a learnable coefficient that controls the relative importance of the interaction between the $p$-th power of the relationship score from modality $a$ and the $q$-th power from modality $b$. This general form is closely related to multilinear polynomials, where the powers of the relationship scores (*i.e.*, $s^p_{(a)i,j}$ and $s^q_{(b)i,j}$) represent the different degrees of interaction between the modalities.

> **Interpretation of multilinear polynomial terms.** The structure of this multilinear polynomial offers valuable insights into the fusion process. Each individual term, such as $s_{(a)i,j}^{p} s_{(b)i,j}^{q}$, represents interactions between higher-order relationships in the graphs $\boldsymbol{G}_{(a)}$ and $\boldsymbol{G}_{(b)}$. These higher-order terms capture increasingly complex dependencies between the two graphs, allowing the model to learn not only from direct pairwise relationships (as seen in the linear terms) but also from more subtle, nonlinear relationships that emerge from specific combinations of scores.

Linear terms like $\boldsymbol{A}_{0,1}s_{(b)i,j}$ or $\boldsymbol{A}_{1,0}s_{(a)i,j}$ represent simple linear combinations of relationship scores from the two graphs. These terms capture first-order interactions, essentially weighting how much each modality's direct relationship contributes to the fused graph. Quadratic and higher-order terms like $\boldsymbol{A}_{1,1}s_{(a)i,j}s_{(b)i,j}$ or $\boldsymbol{A}_{2,2}s_{(a)i,j}^{2}s_{(b)i,j}^{2}$ capture cross-modality interactions that go beyond simple weighting. These terms allow the model to learn relationships in which one modality's relationship score influences the contribution of another modality, enabling more sophisticated fusion strategies. For instance, if two modalities appear only weakly related initially, higher-order terms can help reveal deeper latent connections. This formulation shows that our graph fusion is, in fact, a more general and flexible version of graph-based fusion methods, capable of modeling complex interactions between different modalities. Further insights can be found in Appendices D and E.

## 3.3 MULTI-MODAL VIDEO ANOMALY DETECTION

We present our approach to video anomaly detection for several key reasons:

i. **Multi-modality fusion.** Robust video anomaly detection requires integrating multiple modalities, such as video, audio, and text (Wu et al., 2020; Chen et al., 2023a; Wu et al., 2024). These modalities can be easily obtained, *e.g.*, by using pre-trained video captioning models to generate accompanying text data. For human-related anomaly detection, poses can be extracted using OpenPose (Cao et al., 2017) then embedded into pose features via ST-GCN(Yan et al., 2018). This enables us to explore and evaluate the efficacy of multi-modal feature fusion, where combining complementary information across modalities enhance anomaly detection performance.

ii. **Multi-representational fusion.** Current state-of-the-art video anomaly detection methods typically rely on pre-trained action recognition or motion-based models for feature extraction (Zhu et al., 2024). These models, such as I3D (Carreira & Zisserman, 2017), C3D (Tran et al., 2015) and SwinTransformer (SwinT)(Liu et al., 2022), offer distinct perspectives on the same modality, *e.g.*, videos, extracting different features that represent various aspects of motion, appearance, or temporal dynamics. Our EGO fusion approach is well-suited to this setting, allowing us to combine features from different representations within the same modality, thus enriching the representational capacity of the model and potentially boosting detection accuracy.

iii. **Multi-domain fusion.** Existing video anomaly detection datasets often represent a single domain or scenario, such as videos captured from specific locations like streets or university campuses (Zhu et al., 2024). This limitation offers an opportunity for us to explore multi-domain feature fusion, where features from different environments or scenarios can be integrated to create a more generalized and robust anomaly detection framework. This cross-domain learning can enhance the model's adaptability and performance across diverse settings.

iv. **Binary classification as a foundational task.** Video anomaly detection is commonly framed as a binary classification problem, where the objective is to distinguish between normal and anomalous events. This straightforward approach facilitates intuitive visualization and analysis of the model's performance when implementing our fusion technique, enabling us to gain deeper insights into the impact of feature fusion on detection accuracy. By starting with a binary task, we establish a robust foundational benchmark that allows for iterative refinement and testing of our fusion strategies. Furthermore, once our framework is validated in this context, extending it to more complex multi-class classification tasks becomes a natural next step.

**Degree variance regularization.** Regularization plays a crucial role in enhancing our fused graph representation, which captures the intricate relationships between nodes. In this context, each entry in the fused graph represents the weight of the edge connecting two nodes, providing a quantitative measure of their interconnections. To assess the connectivity of each node, we calculate the sum of the relationship scores for each row in the fused graph. This results in a single value for each node, representing the total weight of all edges linked to it; this value is commonly referred to as the

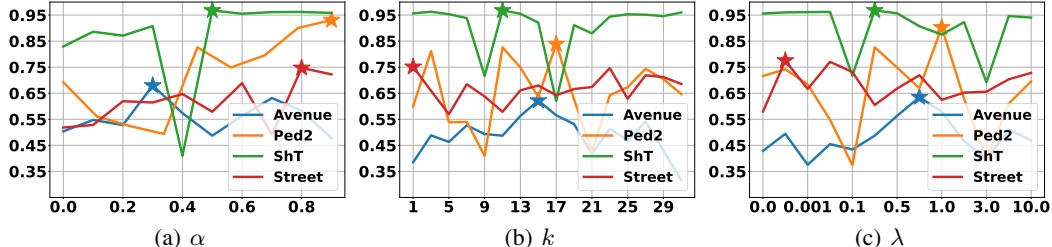

Figure 4: Hyperparameter evaluations for (a) cut-off threshold $\alpha$, (b) top $k$ maximum degrees, and (c) $\lambda$ in the regularization term across all four video anomaly detection datasets, using I3D visual features and text features in our EGO fusion framework.

weighted degree. A weighted degree of zero indicates that a node is isolated with no connections, while a higher weighted degree signifies greater interconnectivity, either through a larger number of edges or edges with higher weights. To facilitate an effective fusion process, we introduce a degree variance regularization term that operates on the fused graph representation for anomaly detection:

$$\ell = \lambda \left\| \text{Var}\left( \sum_{i=1}^{N} \boldsymbol{G}_{i,j}^{-}[s_{i,j} \geq \alpha] \right) - \text{Var}\left( \text{TopMax}_k \left( \sum_{i=1}^{N} \boldsymbol{G}_{i,j}^{+}[s_{i,j} \geq \alpha] \right) \right) \right\|_2^2, \quad (7)$$

where $\lambda$ is a penalty parameter that controls the strength of this regularization, $\boldsymbol{G}^+$ and $\boldsymbol{G}^-$ denote the graphs corresponding to abnormal and normal behaviors, respectively. The parameter $\alpha$ acts as a cut-off threshold, filtering connections such that, for example, $\alpha = 0.5$ excludes relationship scores below this value. Additionally, $\text{TopMax}_k(\cdot)$ selects the top $k$ maximum degree values from the abnormal graph. The purposes of this regularization term is to ensure that unit-level normal features forming nodes have similar degrees in both normal and abnormal graphs.

Our regularization term can be integrated into the original anomaly detection classification loss, such as Binary Cross-Entropy (BCE) loss. By emphasizing the variance in node connectivity, we enhance the model's sensitivity to anomalies while promoting a balanced representation across the graph.

## 4 EXPERIMENTS

### 4.1 SETUP

**Datasets.** We select the following datasets for our evaluation: (i) *UCSD Ped2* (Ped2) features 16 training and 21 testing videos of pedestrians, with anomalies like cyclists, skateboarders, and cars on paths. (ii) *ShanghaiTech* (ShT) has 330 training and 107 testing videos across 13 campus scenes, with 130 abnormal events such as cyclists and fights. (iii) *CUHK Avenue* (Avenue) includes 16 training and 21 testing videos, with 47 anomalies like running, walking in the wrong direction, and object throwing. (iv) *Street Scene* (Street) comprises 46 training and 35 testing videos of a two-lane street, capturing 205 anomalies like jaywalking, U-turns, and car ticketing.

**Features.** We use popular models pretrained on Kinetics-400 (Kay et al., 2017) as feature extractors, including I3D (Carreira & Zisserman, 2017) for 2048-dim. features from each 16-frame segment. We also extract 4096- and 1024-dim. features using pretrained C3D (Tran et al., 2015) and SwinT (Liu et al., 2022), respectively. For text feature extraction, we apply SwinBERT (Lin et al., 2022), pretrained on VATEX (Wang et al., 2019b), to generate dense video captions for every 64-frame segment. We then use SimCSE (Gao et al., 2021) to obtain 768-dim. text embeddings.

**Metrics.** Following common practice (Chen et al., 2023a), we consider Area Under the ROC curve (AUC) which is widely used for evaluation in video anomaly detection. Similar to existing methods like (Tian et al., 2021; Chen et al., 2023b;a), which evaluate frame-level performance by repeating snippet-level predictions (*e.g.*, 16 times) to fit frame-level labels, we adopt a snippet-by-snippet evaluation method as we do not have access to frame-level features. To obtain snippet-level labels, we derive them from the frame-level labels: if any anomaly occurs within a 16-frame snippet, we label the snippet as abnormal; otherwise, it is labeled as normal.

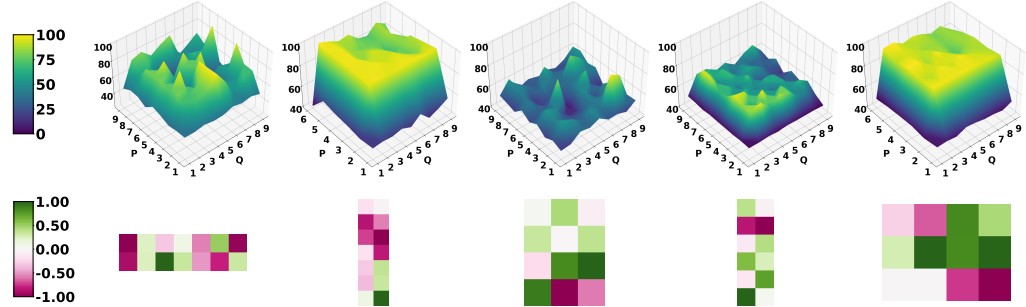

Figure 5: (*Top row*): The effects of $P$ (for visual feature) and $Q$ (for text feature) in the learnable graph operator. (*Bottom row*): The learned optimal $\boldsymbol{A}$ for (*from left to right*) UCSD Ped2, ShanghaiTech, CUHK Avenue, Street Scene, and joint training on both UCSD Ped2 and ShanghaiTech.

Table 1: Experimental results on feature-level and graph-level fusion across four video anomaly detection datasets, including single-modality comparisons. Graph-level single-modality and traditional methods use similarity graph representations for anomaly detection.

| | | UCSD Ped2 | ShanghaiTech | CUHK Avenue | Street Scene |
|---|---|---|---|---|---|
| **Feature-level** | I3D visual | 78.90 | 95.87 | 37.25 | 74.53 |
| | Text only | 80.02 | 83.39 | 65.19 | 69.34 |
| | Concatenation | 86.72 | 96.07 | 43.22 | 75.42 |
| | Addition | 86.20 | 95.77 | 57.44 | 75.05 |
| | Product | 62.72 | 94.15 | 32.04 | 75.59 |
| | MTN fusion | 92.80 | 96.37 | 62.06 | 71.50 |
| **Graph-level** | I3D visual | 68.89 | 69.88 | 58.72 | 49.12 |
| | Text only | 43.03 | 85.59 | 42.36 | 55.27 |
| | Concatenation | 63.45 | 88.68 | 50.09 | 48.97 |
| | Addition | 57.88 | 44.07 | 40.24 | 57.18 |
| | Product | 43.07 | 86.49 | 44.34 | 66.52 |
| | **EGO (ours)** | **93.23** | **97.26** | **83.10** | **77.61** |

**Baselines.** We compare our EGO fusion with both feature- and graph-based approaches, including traditional techniques like simple concatenation and element-wise fusion. We also report single-modality performance for comparison. Additionally, we reproduce the results of a recent multi-modal fusion method (Chen et al., 2023a) to show the effectiveness of our approach. Our classification layer, following the EGO fusion, consists of two fully connected (FC) layers ($N{\rightarrow}N$, $N{\rightarrow}1$) with a ReLU in between, followed by a Sigmoid. For simplicity, we set $N$ to 32. We use cosine similarity to create relationship graphs in our experiments. We set training epochs to 30-50, depending on the datasets; for example, we use 50 for multi-domain experiments with Ped2 and ShT.

## 4.2 EVALUATION

**Discussion on hyperparameter evaluations.** Figure 4 shows our results. First, the cut-off threshold $\alpha$ for filtering weak relationships varies by dataset: higher values (*e.g.*, 0.8 and 0.9) work better for Ped2 and Street, while a lower $\alpha$ is needed for Avenue, likely due to more background motion in the latter. Second, the optimal $k$ for selecting maximum degree values of normal nodes in the abnormal relationship graph is similar across ShT (11), Ped2 (17), and Avenue (15). This similarity may arise from the datasets being captured on campus and featuring comparable anomalies, like cyclists. Interestingly, the optimal $k$ for Street is just 1, which may reflect its complexity due to diverse anomalies in a two-lane street setting. Additionally, the optimal regularization penalty parameter $\lambda$ for Street is low ($1e{-}4$), suggesting a minor effect of regularization on its feature fusion. In contrast, a larger $\lambda$ (*e.g.*, 1) yields the best performance on Ped2.

**A closer look at the learnable graph operator.** We set both $P$ (for I3D visual features) and $Q$ (for SimCSE text embeddings) in equation 5 to range from 1 to 10 and conduct a grid search to evaluate their impact on our EGO fusion. We evaluate the framework on all four individual anomaly detection datasets, as well as on a combination of ShT and Ped2, as shown in Figure 5. The results indicate that $P$ and $Q$ significantly influence EGO fusion performance, suggesting that the fusion

Table 2: Comparison of MTN fusion (feature-level) and EGO fusion (graph-level). ShanghaiTech (ShT) is used for multi-representational and multi-modality fusion, while UCSD Ped2 (Ped2) and ShT are used for multi-domain fusion. Unlike MTN, which fuses two features at a time, EGO fusion enables simultaneous fusion of multiple features for greater flexibility. Training times for one epoch (in seconds) with a batch size of 32 on an Nvidia RTX 4070 GPU are also reported, with model sizes indicated in blue next to their respective models.

| | Train | Test | MTN [29.0M] | | EGO [0.091M] | |
|---|---|---|---|---|---|---|
| | | | AUC | Time | AUC | Time |
| **Multi-represent.** | I3D + C3D | I3D + C3D | **89.25** | 13.6 | 87.17 | **7.8** |
| | I3D + SwinT | I3D + SwinT | 88.80 | 9.7 | **89.85** | **4.9** |
| | C3D + SwinT | C3D + SwinT | 84.45 | 12.0 | **85.52** | **5.7** |
| | I3D + C3D + SwinT | I3D + C3D + SwinT | N/A | - | **95.38** | **9.0** |
| **Multi-modality** | Visual + Text | Visual + Text | 96.37 | | **97.26** | |
| | Visual + Pose | Visual + Pose | 95.48 | | **96.04** | |
| | Text + Pose | Text + Pose | 94.49 | | **95.77** | |
| | Visual + Text + Pose | Visual + Text + Pose | N/A | | **97.79** | |
| **Multi-domain** | Ped2 + ShT | Ped2 only | 56.21 | | **58.30** | |
| | | ShT only | **96.04** | | 95.10 | |
| | | Ped2 + ShT | **94.60** | | 92.11 | |

process is (i) dependent on dataset complexity, (ii) affected by the quality of visual and text features, and (iii) guided by the learning objective, *e.g.*, anomaly detection. By adjusting these parameters, the framework can prune irrelevant connections and strengthen useful ones across different feature modalities. The learned $A$ shows that a higher-order relationship graph is required for visual features in Ped2 and ShT, as well as for their combined dataset. Conversely, text features necessitate a higher-order graph for ShT and Street. This indicates that in more complex scenes, text features can enhance anomaly detection performance.

**Graph-level fusion *vs*. feature-level fusion.** As shown in Table 1, for individual modalities such as visual-only or text-only inputs, raw features typically outperform their corresponding graph representations. This is because raw features are high-dimensional (*e.g.*, I3D visual features are 2048-dimensional, and text features are 768-dimensional), containing rich semantic information directly, whereas graph representations primarily capture relationships and are much lower-dimensional. We also observe that feature-level fusion techniques, such as concatenation, addition, and product, tend to enhance performance compared to using individual feature modalities. However, these same methods do not perform as well when applied to graph-level fusion, indicating the need for more sophisticated fusion strategies at the graph level. Additionally, we compare our approach to the popular Multi-scale Temporal Network (MTN) fusion (Chen et al., 2023a), which fuses visual and text features in an end-to-end learnable manner at the feature level. Our EGO fusion consistently outperforms MTN fusion on all benchmarks while being more lightweight and interpretable.

**Unified fusion across representations, modalities, and domains.** Table 2 summarizes our experiments on multi-representational, multi-modal, and multi-domain feature fusion. As shown, our EGO fusion outperforms the MTN fusion (Chen et al., 2023a), despite its simplicity. A key advantage of EGO fusion is its ability to integrate multiple modalities, representations, and domains simultaneously, whereas MTN is limited by its network design to fusing only two features at a time. Moreover, our framework is more lightweight and significantly faster to train.

## 5 CONCLUSION

In this paper, we introduce a novel graph-based fusion framework, termed EGO fusion, which enhances feature integration. By using relationship graphs on raw features, our approach captures deeper interactions through iterative graph relationship updates and a learnable fusion operator. Our framework not only reduces dimensionality and computational costs but also improves interpretability by emphasizing feature relationships. Experiments in video anomaly detection show that our method outperforms traditional fusion techniques, highlighting the potential of relationship-driven fusion approaches in using multi-representational, multi-modal, and multi-domain features. We believe our contributions will inspire further research into understanding complex feature interactions, ultimately leading to more robust model performance across diverse applications.

ACKNOWLEDGMENTS

Dexuan Ding conducted this research under the supervision of Lei Wang as part of his final year honors project at ANU. We extend our gratitude to Qixiang Chen for reviewing the model code and producing excellent plots. This work was also supported by the NCI National AI Flagship Merit Allocation Scheme, and the National Computational Merit Allocation Scheme 2024 (NCMAS 2024), with computational resources provided by NCI Australia, an NCRIS-enabled capability supported by the Australian Government.

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

## A  UNIT-LEVEL REPRESENTATION

> **Unit-level** refers to the fundamental, granular components of data that can be segmented and analyzed independently. Depending on the data modality, unit-level entities include words or paragraphs in text, patches in images, frames, clips, or cubes in videos, and tokens in sequential data. These units represent the smallest meaningful segments of the data, serving as essential building blocks for deeper analysis and processing.

> **Unit-level features** are the extracted representations of these granular data components. Derived from individual units, such as words or paragraphs in text, patches in images, frames, cubes, or clips in videos, or tokens in sequences, unit-level features capture the distinctive characteristics of each unit. These features are widely used in recent advances like large language models (LLMs) and vision-language models (VLMs), where the ability to capture fine-grained details at the unit level has significantly improved tasks such as text generation, image classification, object recognition, and multimodal understanding. By using unit-level features, modern models achieve superior performance in higher-level tasks, including classification, recognition, prediction, and even cross-modal tasks, where detailed analysis is crucial for accurate and contextually rich outcomes.

## B  RELATIONSHIP BETWEEN FEATURE DISTANCE AND SIMILARITY

A widely used feature distance measure is the Euclidean distance. When we use Euclidean distance to measure the similarity between two network-encoded, $L^2$-normalized features from two images, we obtain the following expression:

$$
\begin{aligned}
||\phi(\boldsymbol{X}) - \phi(\boldsymbol{Y})||_2^2 &= \langle \phi(\boldsymbol{X}), \phi(\boldsymbol{X}) \rangle - 2\langle \phi(\boldsymbol{X}), \phi(\boldsymbol{Y}) \rangle + \langle \phi(\boldsymbol{Y}), \phi(\boldsymbol{Y}) \rangle \\
&= 2 - 2\langle \phi(\boldsymbol{X}), \phi(\boldsymbol{Y}) \rangle \\
&\equiv 2 - 2k(\phi(\boldsymbol{X}), \phi(\boldsymbol{Y}))
\end{aligned}
\tag{8}
$$

In this equation, $\phi(\boldsymbol{X})$ and $\phi(\boldsymbol{Y})$ represent the feature maps of images $\boldsymbol{X}$ and $\boldsymbol{Y}$, respectively, while $k(\cdot, \cdot)$ denotes various types of similarity measures. These measures can include dynamic time warping (DTW)(Cuturi, 2011) and its variants such as soft-DTW(Cuturi & Blondel, 2017), uncertainty-DTW (Wang & Koniusz, 2022b), and JEANIE (Wang et al., 2021; Wang & Koniusz, 2022a; Wang et al., 2024), or kernels such as intersection and radial basis function (RBF) kernels, as well as simpler metrics like cosine similarity. Since the features $\phi(\boldsymbol{X})$ and $\phi(\boldsymbol{Y})$ are $L^2$-normalized, both $k(\phi(\boldsymbol{X}), \phi(\boldsymbol{X}))$ and $k(\phi(\boldsymbol{Y}), \phi(\boldsymbol{Y}))$ equal 1.

> This shows that the Euclidean distance between two $L^2$-normalized features is directly related to the similarity measure between them. Specifically, the Euclidean distance can be expressed in terms of the similarity function $k(\cdot, \cdot)$, where higher similarity, as measured by $k(\phi(\boldsymbol{X}), \phi(\boldsymbol{Y}))$, leads to a smaller Euclidean distance. In the case where the features are identical, the similarity reaches its maximum value of 1, and the Euclidean distance is zero. Conversely, as the similarity decreases, the distance increases. Thus, this formulation highlights the inverse relationship between distance and similarity for normalized features, emphasizing that both metrics are fundamentally tied to how well the features align.

## C  DERIVATION OF GRAPH FUSION OPERATOR IN EGO FUSION

We begin by applying learnable weights, denoted as $\boldsymbol{a}$ and $\boldsymbol{b}$, to two iterative graph relationship updates $\mathbf{G}_{(a)}$ and $\hat{\mathbf{G}}_{(b)}$. These vectors learn to select graph relationship updates for the fusion of two

graphs. The process can be expressed as follows:

$$
\boldsymbol{G} = \mathbf{G}_{(a)} \circledast \boldsymbol{A} \circledast \mathbf{G}_{(b)}^{\mathsf{T}} = \sum_{q=0}^{Q}\sum_{p=0}^{P} \boldsymbol{R}_{(a)}^{p} a_P \odot \boldsymbol{R}_{(b)}^{q} b_q = \sum_{q=0}^{Q}\sum_{p=0}^{P} a_p b_q \left( \boldsymbol{R}_{(a)}^{p} \odot \boldsymbol{R}_{(b)}^{q} \right)
$$

$$
= \sum_{q=0}^{Q}\sum_{p=0}^{P} (\boldsymbol{a} \otimes \boldsymbol{b}) \odot \left( \mathbf{G}_{(a)} \circledcirc \mathbf{G}_{(b)} \right)
$$

$$
= \sum_{q=0}^{Q}\sum_{p=0}^{P} \left( \begin{bmatrix} a_0 \\ a_1 \\ \vdots \\ a_P \end{bmatrix} \otimes [b_0, b_1, \cdots, b_Q] \right) \odot \left( \begin{bmatrix} \boldsymbol{R}_{(a)}^{0} \\ \boldsymbol{R}_{(a)}^{1} \\ \vdots \\ \boldsymbol{R}_{(a)}^{P} \end{bmatrix} \circledcirc \left[ \boldsymbol{R}_{(b)}^{0}, \boldsymbol{R}_{(b)}^{1}, \cdots, \boldsymbol{R}_{(b)}^{Q} \right] \right)
$$

$$
= \sum_{q=0}^{Q}\sum_{p=0}^{P} \begin{bmatrix} a_0 b_0 & a_0 b_1 & a_0 b_2 & \dots & a_0 b_Q \\ a_1 b_0 & a_1 b_1 & a_1 b_2 & \dots & a_1 b_Q \\ \vdots & \vdots & \vdots & \ddots & \vdots \\ a_P b_0 & a_P b_1 & a_P b_2 & \dots & a_P b_Q \end{bmatrix} \odot \begin{bmatrix} \boldsymbol{R}_{(a)}^{0} \odot \boldsymbol{R}_{(b)}^{0} & \boldsymbol{R}_{(a)}^{0} \odot \boldsymbol{R}_{(b)}^{1} & \boldsymbol{R}_{(a)}^{0} \odot \boldsymbol{R}_{(b)}^{2} & \dots & \boldsymbol{R}_{(a)}^{0} \odot \boldsymbol{R}_{(b)}^{Q} \\ \boldsymbol{R}_{(a)}^{1} \odot \boldsymbol{R}_{(b)}^{0} & \boldsymbol{R}_{(a)}^{1} \odot \boldsymbol{R}_{(b)}^{1} & \boldsymbol{R}_{(a)}^{1} \odot \boldsymbol{R}_{(b)}^{2} & \dots & \boldsymbol{R}_{(a)}^{1} \odot \boldsymbol{R}_{(b)}^{Q} \\ \vdots & \vdots & \vdots & \ddots & \vdots \\ \boldsymbol{R}_{(a)}^{P} \odot \boldsymbol{R}_{(b)}^{0} & \boldsymbol{R}_{(a)}^{P} \odot \boldsymbol{R}_{(b)}^{1} & \boldsymbol{R}_{(a)}^{P} \odot \boldsymbol{R}_{(b)}^{2} & \dots & \boldsymbol{R}_{(a)}^{P} \odot \boldsymbol{R}_{(b)}^{Q} \end{bmatrix}
$$

$$
= \sum_{q=0}^{Q}\sum_{p=0}^{P} \boldsymbol{A} \odot \begin{bmatrix} \boldsymbol{R}_{(a)}^{0} \odot \boldsymbol{R}_{(b)}^{0} & \boldsymbol{R}_{(a)}^{0} \odot \boldsymbol{R}_{(b)}^{1} & \boldsymbol{R}_{(a)}^{0} \odot \boldsymbol{R}_{(b)}^{2} & \cdots & \boldsymbol{R}_{(a)}^{0} \odot \boldsymbol{R}_{(b)}^{Q} \\ \boldsymbol{R}_{(a)}^{1} \odot \boldsymbol{R}_{(b)}^{0} & \boldsymbol{R}_{(a)}^{1} \odot \boldsymbol{R}_{(b)}^{1} & \boldsymbol{R}_{(a)}^{1} \odot \boldsymbol{R}_{(b)}^{2} & \cdots & \boldsymbol{R}_{(a)}^{1} \odot \boldsymbol{R}_{(b)}^{Q} \\ \vdots & \vdots & \vdots & \ddots & \vdots \\ \boldsymbol{R}_{(a)}^{P} \odot \boldsymbol{R}_{(b)}^{0} & \boldsymbol{R}_{(a)}^{P} \odot \boldsymbol{R}_{(b)}^{1} & \boldsymbol{R}_{(a)}^{P} \odot \boldsymbol{R}_{(b)}^{2} & \cdots & \boldsymbol{R}_{(a)}^{P} \odot \boldsymbol{R}_{(b)}^{Q} . \end{bmatrix} \tag{9}
$$

In this formulation, $\otimes$ represents the outer product, $\odot$ denotes element-wise multiplication, and $\circledcirc$ is an operation we define that functions similarly to the outer product but uses element-wise multiplication for the fusion process.

As shown in equation 9, instead of using two independent vectors $\boldsymbol{a}$ and $\boldsymbol{b}$ that result in $\boldsymbol{A} = \boldsymbol{a} \otimes \boldsymbol{b}$, we can make $\boldsymbol{A}$ a fully learnable matrix. This allows the model to explore a more efficient and flexible fusion process by learning the weights directly, enabling more effective integration of information from both graphs.

## D  NONLINEAR RELATIONSHIPS AND EXPRESSIVITY

The ability to model these nonlinear relationships through multilinear polynomials provides significant expressivity for graph fusion. By including powers of the similarity scores, the fused graph can effectively capture subtle nuances in the relationships between the modalities that a purely linear model might miss. This is especially important in cases where different modalities carry complementary information. For example:

**Complementary modality information.** Consider video data with accompanying textual descriptions. The visual modality might capture a general scene, while the text might provide specific context (*e.g.*, objects or actions). Linear combinations of similarities might only highlight directly overlapping information, but higher-order polynomial terms can reveal deeper, context-driven relationships, such as when visual cues and descriptive language align only in specific scenarios.

**Disentangling complex correlations.** Higher-order terms also help disentangle complex correlations between modalities. For instance, a quadratic term might reveal situations where both modalities strongly correlate with a particular feature (*e.g.*, object presence in an image and its mention in the text) only when viewed together, even if individually their similarities are weak. The polynomial structure allows the fusion operator to amplify such interactions.

## E  LEARNABLE WEIGHTS AND OPTIMIZATION

The learnable coefficients $\boldsymbol{A}_{p,q}$ ($p \in \mathcal{I}_P$, $q \in \mathcal{I}_Q$) play a crucial role in determining how the contributions from different terms are weighted. During training, the model optimizes these coefficients

through backpropagation to balance the linear and nonlinear terms in the multilinear polynomial. This enables the model to learn the most relevant interactions for the task at hand, whether they be direct similarities (captured by lower-order terms) or more complex, indirect relationships (captured by higher-order terms).

**Multilinear fusion in practice.** In practical applications, the multilinear polynomial fusion approach offers several benefits: (i) Adaptability: The fusion process can adapt to different levels of interaction between modalities, adjusting the contribution of each term based on the complexity of the task. (ii) Robustness: By incorporating both linear and nonlinear terms, the fused graph is more robust to noisy or missing data. Even if one modality's similarity is weak, higher-order terms involving the other modality can still provide meaningful information. (iii) Improved task performance: Tasks such as classification, retrieval, or multi-modal feature learning can benefit from this fusion strategy, as it enables the model to use both direct and subtle modality interactions for better decision-making.

## F    OPTIMAL HYPERPARAMETERS FOR EACH DATASET

Table 3: Optimal hyperparameters for each dataset.

| Dataset | Operator | $m$ | $n$ | $\lambda$ | $\alpha$ | $k$ | Best AUC |
|---|---|---|---|---|---|---|---|
| CUHK Avenue | $\boldsymbol{a} \otimes \boldsymbol{b}$ | 2 | 7 | 1 | 0.5 | 10 | 83.10 |
| ShanghaiTech (ShT) | $\boldsymbol{A}$ | 2 | 6 | 1 | 0.5 | 10 | 97.26 |
| UCSD Ped2 (Ped2) | $\boldsymbol{A}$ | 4 | 3 | 1 | 0.5 | 10 | 93.23 |
| Street Scene | $\boldsymbol{A}$ | 4 | 3 | 1 | 0.5 | 10 | 77.61 |
| Combined (ShT + Ped2) | $\boldsymbol{A}$ | 4 | 4 | 0.001 | 0.5 | 10 | 92.88 |

Selecting the appropriate hyperparameters is critical for achieving optimal performance across different datasets. As shown in Table 3, we determine the best hyperparameters by conducting a grid search across five datasets, including one combined dataset (ShanghaiTech and UCSD Ped2). The fusion operator corresponds to two distinct learnable weight matrix representations: it can either be modeled as a matrix $\boldsymbol{A}$ (see equation 5) or as the outer product of $\boldsymbol{a}$ and $\boldsymbol{b}$ (see equation 4). The remaining hyperparameters are fine-tuned through grid search to ensure the best performance.

## G    RELATIONSHIP BETWEEN EGO AND ITS REAL-WORLD APPLICATIONS

Iterative graph relationship updates play a crucial role in modeling complex interactions within graphs, enabling a more refined and adaptive fusion of information. Unlike traditional methods that rely on direct connections, our approach incrementally enhances relationships through element-wise operations, allowing for a more controlled and localized expansion of graph structure.

In video anomaly detection, iterative updates uncover subtle dependencies between frames by continuously refining relationships over multiple iterations. This process improves the detection of anomalous behaviors that single-frame analysis might overlook by capturing evolving contextual cues.

For multi-modal data fusion, iterative graph relationship updates facilitate seamless alignment between different modalities. For example, in image-captioning tasks, textual descriptions may reference background elements indirectly. Our approach adaptively strengthens such implicit connections, improving the integration of visual and textual representations. Similarly, in social network analysis, iterative updates refine the understanding of information diffusion and influence chains. Instead of relying on predefined connectivity patterns, this approach dynamically adjusts relationships, better capturing the evolving nature of interactions across a network.

By expanding graph relationships in an iterative and structured manner, our method provides a more flexible and effective framework for capturing nuanced dependencies across various domains.

Table 4: Experimental results on XD-Violence and UCF-Crime.

|  |  | XD-Violence (AP) | UCF-Crime (AUC) |
|---|---|---|---|
| **Feature-level** | I3D visual | 60.96 | 76.02 |
|  | Text only | 51.31 | 69.23 |
|  | Concat. | 59.93 | 67.84 |
|  | Addition | 58.79 | 68.93 |
|  | Product | 24.10 | 50.23 |
|  | MTN fusion | 77.17 | 85.14 |
| **Graph-level** | I3D visual | 34.14 | 61.57 |
|  | Text only | 24.42 | 58.72 |
|  | Concat. | 27.89 | 63.16 |
|  | Addition | 27.66 | 65.30 |
|  | Product | 24.10 | 50.23 |
|  | **EGO (ours)** | 65.77 | 81.71 |

Table 5: Experimental results on the Multi-Scenario Anomaly Detection (MSAD) Dataset.

|  | Venue | MSAD |
|---|---|---|
| MIST (I3D) (Feng et al., 2021) | ICCV 2021 | 86.65 |
| MIST (SwinT) (Feng et al., 2021) | ICCV 2021 | 85.67 |
| UR-DMU (Zhou et al., 2023) | AAAI 2023 | 85.02 |
| UR-DMU (SwinT) (Zhou et al., 2023) | AAAI 2023 | 72.36 |
| MGFN (I3D)(Chen et al., 2023b) | AAAI 2023 | 84.96 |
| MGFN (SwinT) (Chen et al., 2023b) | AAAI 2023 | 78.94 |
| MTN (I3D)(Chen et al., 2023a) | CVPRW2023 | 86.82 |
| MTN (SwinT) (Chen et al., 2023a) | CVPRW2023 | 83.60 |
| **EGO (ours)** | - | **87.36** |

## H    EGO WITH EXISTING FRAMEWORKS AND PIPELINES

EGO Fusion is designed with modularity at its core, ensuring compatibility with both (i) widely used machine learning frameworks, such as PyTorch and TensorFlow, and (ii) diverse model architectures. By using standard adjacency matrices and similarity metrics, EGO Fusion easily integrates with pre-trained models and raw feature pipelines. It can be seamlessly integrated with video backbones such as C3D, I3D, SwinTransformer, as well as GCN-based models for human skeleton sequences in pose modality (as demonstrated in Table 2). Additionally, EGO Fusion supports text embedding models, including SimCSE.

## I    SCALABILITY OF EGO FOR LARGE-SCALE DATASETS

**Computational complexity.** EGO graphs are inherently small, such as $32 \times 32$ adjacency matrices when a video is divided into $N = 32$ video clips (a.k.a. temporal blocks). Computing matrix-matrix multiplications with a non-parallelized algorithm has a complexity of $\mathcal{O}(N^{2.37})$ (Le Gall, 2014). On GPUs (parallel hardware), this complexity is significantly reduced to $\mathcal{O}(\log(N))$. Consequently, the dominant computational complexity arises from equation 3, which is $\mathcal{O}((P+Q)\log(N))$. Element-wise multiplications in equation 4 are highly parallelizable, making their complexity negligible for typical values of $P = Q \approx 8$.

EGO Fusion is thus highly scalable, using lower-dimensional relationship graph spaces to reduce computational overhead compared to feature-level (high-dimensional) fusion methods. Further-more, the learnable graph operator effectively balances self-connections with iterative graph rela-tionship updates. As shown in Table 2, EGO reduces training times by up to 50% compared to MTN on the ShanghaiTech dataset. For real-time applications, the learnable graph operator selec-tively refines key relationships to ensure computational efficiency. Additionally, sparse matrix (fast) representations can be used for further performance optimization.

Table 6: Experimental results by anomaly type (11 main anomaly types) on the MSAD dataset.

| Method | Assault | | Explosion | | Fighting | | Fire | |
|---|---|---|---|---|---|---|---|---|
| | AUC | AP | AUC | AP | AUC | AP | AUC | AP |
| RTFM (Tian et al., 2021) | **68.1** | **67.3** | 46.8 | 60.4 | **89.6** | 93.0 | 61.3 | 81.2 |
| MGFN (Chen et al., 2023b) | 59.7 | 59.0 | 64.5 | 71.9 | 89.4 | 93.5 | **86.0** | 93.0 |
| UR-DMU (Zhou et al., 2023) | 56.9 | 64.5 | **67.9** | **74.5** | 83.9 | 90.4 | 61.2 | 82.9 |
| **EGO (Ours)** | 52.2 | 57.5 | 57.6 | 74.4 | 66.5 | 72.8 | 62.9 | **86.7** |

| Method | Object Falling | | People Falling | | Robbery | | Shooting | |
|---|---|---|---|---|---|---|---|---|
| | AUC | AP | AUC | AP | AUC | AP | AUC | AP |
| RTFM (Tian et al., 2021) | **94.7** | **96.7** | **56.5** | **50.4** | 65.7 | 81.2 | 78.2 | 84.7 |
| MGFN (Chen et al., 2023b) | 90.9 | 94.8 | 52.7 | 47.8 | **73.9** | 86.7 | **86.8** | **88.5** |
| UR-DMU (Zhou et al., 2023) | 92.1 | 95.8 | 42.5 | 43.7 | 63.5 | 79.3 | 81.4 | 87.8 |
| **EGO (Ours)** | 92.3 | 94.8 | 35.4 | 43.8 | 64.8 | **87.5** | 68.6 | 78.4 |

| Method | Traffic Accident | | Vandalism | | Water Incident | | **Overall** | |
|---|---|---|---|---|---|---|---|---|
| | AUC | AP | AUC | AP | AUC | AP | AUC | AP |
| RTFM (Tian et al., 2021) | 62.2 | 51.8 | 85.2 | 76.1 | **96.3** | 99.1 | 86.7 | 66.3 |
| MGFN (Chen et al., 2023b) | 68.6 | 54.5 | 82.4 | 80.1 | 85.5 | 97.0 | 85.0 | 63.5 |
| UR-DMU (Zhou et al., 2023) | 62.0 | 55.6 | 84.7 | 77.0 | 98.5 | **99.5** | 85.0 | **68.3** |
| **EGO (Ours)** | **69.9** | **64.3** | **88.1** | **81.4** | 81.9 | 95.4 | **87.3** | 64.4 |

**Datasets.** We evaluate EGO on large-scale anomaly detection datasets and have selected the following: (i) *XD-Violence*: This dataset contains 3,954 training videos and 800 testing videos across various scenes, primarily from games or movies. It features anomalies such as abuse, explosions, fighting, and riots. (ii) *UCF-Crime*: This dataset includes 1,610 training videos and 290 testing videos of real-world multi-scene surveillance footage. It encompasses 13 types of anomalies, including fires, fights, and robberies. (iii) Multi-Scenario Anomaly Detection (MSAD): Introduced by Zhu et al. (2024), this dataset consists of 360 training videos and 360 testing videos (Protocol ii) from multi-scenario surveillance footage. It contains 55 types of anomalies, including assaults, explosions, and people falling.

**Setups.** For visual feature extraction, we use I3D pretrained on Kinetics-400 to obtain 2048-dimensional features. For text feature extraction, we use SimCSE (as described in Sec. 4.1) to generate 768-dimensional text embeddings. On the MSAD dataset, we additionally extract 1024-dimensional features using pretrained I3D and SwinTransformer (SwinT), respectively. Below, we show evaluations on three large-scale anomaly detection datasets. We also present an evaluation by scenario (14 scenarios) on MSAD, and the results (AUC) follow the Protocol ii of (Zhu et al., 2024).

**Evaluations.** As shown in the large-scale evaluations (Table 4, 5, 6, and 7), we draw the following insights: (i) EGO maintains high efficiency on large and complex datasets. (ii) Compared to recent deep learning models, such as the latest MTN fusion (Chen et al., 2023a), our EGO fusion demonstrates robustness in multi-scenario anomaly detection for static surveillance data. This is attributed to the ability of our relationship graph to effectively capture feature relationships, enabling the modeling of richer and more nuanced structural interactions that conventional fusion methods often overlook. (iii) EGO performs well on UCF-Crime videos, even in cases where the footage includes frequently changing camera viewpoints. (iv) EGO is designed with computational efficiency in mind, using only 0.091M parameters, significantly fewer than MTN (which uses 29.0M parameters). This reduction in parameters results in lower memory usage and faster inference times. Note that the weighted sums in EGO Fusion are dynamically learned through a task-specific optimization process, allowing for flexible, data-driven adaptation to the unique complexities of each dataset.

**Discussions.** Existing cross-modal graph methods (Yu et al., 2022; He et al., 2021) primarily rely on static inter-graph convolutions. In contrast, EGO captures higher-order relationships through iterative graph relationship updates: (i) We integrate both direct and iteratively refined connections. (ii) Equation 6 shows that the adjacency relationships between modalities are aligned via attention, *i.e.*, $\boldsymbol{G} = \sum_p \sum_q \boldsymbol{A}_{pq}(\boldsymbol{R}^p_{\text{modality1}} \odot \boldsymbol{R}^q_{\text{modality2}})$. (iii) Our polynomial expansion scheme accounts for the fact that latent correlations within each modality may vary dynamically. Additionally, our

Table 7: Experimental results by scenario (14 total scenarios) on the MSAD dataset.

| Method | Frontdoor | | Highway | | Mall | | Office | |
|---|---|---|---|---|---|---|---|---|
| | AUC | AP | AUC | AP | AUC | AP | AUC | AP |
| RTFM (Tian et al., 2021) | 84.1 | 81.1 | 63.7 | 4.1 | 87.2 | 72.2 | 78.1 | 68.8 |
| MGFN (Chen et al., 2023b) | **86.4** | **85.1** | 79.7 | 4.1 | 65.3 | 56.6 | 75.1 | 62.4 |
| UR-DMU (Zhou et al., 2023) | 84.8 | 82.8 | 31.5 | 1.3 | **91.0** | **83.8** | 77.8 | 67.3 |
| **EGO (Ours)** | 85.2 | 81.6 | **80.2** | **30.8** | 82.3 | 73.4 | **80.0** | **71.7** |

| Method | Park | | Parkinglot | | Pedestrian st. | | Restaurant | |
|---|---|---|---|---|---|---|---|---|
| | AUC | AP | AUC | AP | AUC | AP | AUC | AP |
| RTFM (Tian et al., 2021) | 69.0 | 25.6 | 74.4 | 35.9 | 97.4 | 50.6 | **96.1** | **91.9** |
| MGFN (Chen et al., 2023b) | 77.9 | 38.3 | 68.1 | 14.5 | 88.0 | 20.4 | 95.8 | 91.8 |
| UR-DMU (Zhou et al., 2023) | 87.8 | 36.2 | 91.4 | 53.9 | 81.9 | 11.5 | 93.1 | 87.4 |
| **EGO (Ours)** | **93.5** | **44.3** | **96.8** | **75.2** | **97.5** | **52.0** | 94.3 | 73.9 |

| Method | Road | | Shop | | Sidewalk | | Street highview | |
|---|---|---|---|---|---|---|---|---|
| | AUC | AP | AUC | AP | AUC | AP | AUC | AP |
| RTFM (Tian et al., 2021) | 54.0 | 16.8 | 80.6 | **77.3** | 52.5 | 17.1 | 43.3 | 12.3 |
| MGFN (Chen et al., 2023b) | 77.9 | 49.7 | **84.9** | 77.2 | 85.5 | 62.3 | **87.6** | **40.7** |
| UR-DMU (Zhou et al., 2023) | 83.0 | 64.4 | 81.3 | 64.5 | 86.5 | **64.1** | 85.0 | 37.7 |
| **EGO (Ours)** | **89.8** | **64.6** | 83.4 | 72.2 | **87.1** | 45.0 | 28.2 | 10.1 |

| Method | Train | | Warehouse | | **Overall** | |
|---|---|---|---|---|---|---|
| | AUC | AP | AUC | AP | AUC | AP |
| RTFM (Tian et al., 2021) | 66.9 | 3.9 | 69.5 | 37.4 | 86.7 | 66.3 |
| MGFN (Chen et al., 2023b) | 53.0 | 3.1 | 72.3 | 30.9 | 85.0 | 63.5 |
| UR-DMU (Zhou et al., 2023) | 59.0 | 3.1 | 81.2 | **59.1** | 85.0 | **68.3** |
| **EGO (Ours)** | **80.8** | **7.8** | **84.7** | 46.6 | **87.3** | 64.4 |

variance regularization loss in equation 7 compares the polynomial-aggregated variances of regular and anomalous $G$, a novel aspect of our approach. In contrast, methods such as (Yu et al., 2022; He et al., 2021) use fixed graphs and cross-attention graphs, respectively. The approach in (He et al., 2021) also relies on feature reconstructor losses per modality and triplet loss. Thus, these architectures differ fundamentally from ours. Note that (Yu et al., 2022; He et al., 2021) are designed for remote sensing retrieval and image-text retrieval, whereas EGO is tailored for anomaly detection.

This richer interaction is reflected in our experimental results, where EGO consistently outperforms baseline methods, including state-of-the-art cross-modal approaches, in anomaly detection tasks. Specifically, compared with other anomaly detection models: (i) Our EGO fusion consists of only two fully connected (FC) layers with a ReLU activation in between, followed by a Sigmoid activation. (ii) In contrast, attention-based fusion operates at the feature level, which can be noisy. (iii) By operating at the graph level, our approach is more robust, as demonstrated in the results above.

Additionally, attention-based fusion (He et al., 2021) typically requires three projection layers (for query, key, and value), leading to a significantly higher number of learnable parameters. Their fusion mechanisms are often constrained to a single modality, such as the self-attention mechanisms used in models like MGFN (Chen et al., 2023b) and UR-DMU (Zhou et al., 2023). In contrast, our EGO fusion demonstrates strong performance across multi-representational, multi-modal, and multi-domain feature fusion tasks.

## J  COMPARISON OF TRAINING AND TESTING TIMES

We compare the training and testing times per video across six datasets using both the EGO fusion and the recent MTN fusion approaches. All experiments are conducted on a single Nvidia V100 GPU with a batch size of 32. The training and testing times are measured at the sample level, representing the time required to process a single video.

Table 8: Comparison of training and testing times for EGO and MTN fusion across six datasets.

| | EGO: Training (s) | EGO: Testing (s) | MTN: Training (s) | MTN: Testing (s) |
|---|---|---|---|---|
| UCSD Ped2 | 0.0878 ± 0.0043 | 0.0033 ± 0.0030 | 0.3656 ± 0.0137 | 0.1256 ± 0.0226 |
| ShanghaiTech | 0.1537 ± 0.0160 | 0.0077 ± 0.0041 | 0.4481 ± 0.0232 | 0.1479 ± 0.0127 |
| CUHK Avenue | 0.1472 ± 0.0211 | 0.0078 ± 0.0058 | 0.4338 ± 0.0518 | 0.1384 ± 0.0352 |
| Street Scene | 0.2064 ± 0.0270 | 0.0173 ± 0.0067 | 0.6179 ± 0.0654 | 0.1596 ± 0.0199 |
| XD-Violence | 0.2151 ± 0.0440 | 0.0196 ± 0.0089 | 0.5135 ± 0.0361 | 0.1523 ± 0.0122 |
| UCF-Crime | 0.2427 ± 0.0361 | 0.0195 ± 0.0071 | 0.6484 ± 0.0492 | 0.1699 ± 0.0257 |

Table 9: Performance of EGO in visual and text fusion under varying noise conditions on text features using the ShanghaiTech dataset. We report the Area Under the Curve (AUC).

| Condition | Original | 10% Noise | 30% Noise | 50% Noise |
|---|---|---|---|---|
| Train on Noisy, Test on Clean | 97.26 | 96.01 | 95.98 | 95.58 |
| Train on Clean, Test on Noisy | 97.26 | 95.96 | 95.86 | 95.62 |
| Train on Noisy, Test on Noisy | 97.26 | 95.92 | 95.76 | 94.86 |

As shown in Table 8, EGO fusion achieves significantly lower training and testing times compared to MTN fusion, while delivering comparable or even superior performance in terms of results.

## K  ROBUSTNESS AND CROSS-DATASET GENERALIZATION OF EGO FUSION

**Handling noisy features.** EGO effectively addresses noisy or irrelevant features in the input data through its degree variance regularization mechanism (equation 7). This approach promotes sparsity by diminishing the influence of weak or irrelevant connections in the fused graph, thereby helping to isolate and preserve meaningful relationships.

To evaluate EGO's performance under noisy conditions, we add Gaussian noise to the text features in the ShanghaiTech dataset. The results are presented in Table 9, with the noise ratio indicated as a percentage.

Our findings show that, in the presence of noise, it becomes more challenging to construct a meaningful relationship graph, as the connections between features weaken with increasing noise. Nevertheless, EGO maintains high performance, with accuracy remaining within 2% of the baseline (Original) across all noise levels. This demonstrates EGO's ability to effectively isolate relevant features and minimize the impact of noise, even when the relationship graph is less reliable. The degree variance regularization enables EGO to prioritize stronger, more meaningful connections while de-emphasizing weaker or irrelevant ones. This mechanism enhances the model's robustness in noisy environments, ensuring that noise has minimal impact on overall performance.

**Non-robust feature extraction models.** While high-quality features typically improve performance, EGO Fusion effectively mitigates the impact of less robust inputs by focusing on the relative relationships between features rather than their direct quality. Even when suboptimal features from lower-capacity models are used, EGO's regularization mechanisms and learnable fusion operator help maintain stable performance.

Table 10: EGO performance on different feature combinations. We report the Area Under the Curve (AUC).

| Feature Combination | EGO |
|---|---|
| I3D + SwinT | 89.85 |
| I3D + C3D | 87.17 |
| SwinT + C3D | 85.52 |
| I3D + SwinT + C3D | 95.38 |

Table 11: Comparison of MTN fusion and EGO fusion performance in cross-dataset evaluation. Both models are trained on the ShanghaiTech dataset and evaluated on the UCSD Ped2, CUHK Avenue, Street Scene, XD-Violence, and UCF-Crime datasets. We report the Area Under the Curve (AUC).

| Dataset | UCSD Ped2 | CUHK Avenue | Street Scene | XD-Violence | UCF-Crime |
|---|---|---|---|---|---|
| MTN fusion | 50.49 | 46.99 | 28.94 | 29.65 | 35.08 |
| **EGO (ours)** | 48.03 | 49.35 | 36.76 | 30.52 | 57.84 |

As shown in (Zhu et al., 2024), I3D and SwinT features generally outperform C3D features, indicating that C3D features are less robust. Below, we evaluate EGO's performance when fusing C3D features on the ShanghaiTech dataset, with results provided in Table 10. These results suggest that fusing two features, one of which is less robust, leads to a slight drop in performance. For instance, I3D + SwinT outperforms I3D + C3D by more than 2%, and SwinT + I3D outperforms SwinT + C3D by more than 4%. However, when fusing all three features (I3D + SwinT + C3D), performance improves compared to any pairwise combination. These findings highlight EGO's robustness to variability in feature extraction quality.

**Cross-dataset evaluation.** In this section, we evaluate the performance of EGO fusion and MTN fusion using I3D visual features and text features, as described in Section 4.1. The models are trained on the ShanghaiTech dataset and tested across several other datasets, including UCSD Ped2 (Ped2), CUHK Avenue (Avenue), Street Scene (Street), XD-Violence, and UCF-Crime.

As shown in Table 11, EGO fusion demonstrates strong performance in cross-dataset evaluations. Specifically: (i) Cross-dataset generalization: Despite being trained on the ShanghaiTech dataset, EGO fusion achieves competitive results on other datasets, such as 48.03 on Ped2 and 49.35 on Avenue. (ii) Diverse scenarios: The results on datasets with varying motion patterns and complexities, such as XD-Violence (30.52) and UCF-Crime (57.84), highlight EGO fusion's ability to adapt to datasets with diverse domain characteristics.

These results highlight EGO fusion's strong ability to generalize beyond the training dataset, consistently performing well across datasets with varying characteristics and challenges.

## L  ADDITIONAL VISUALISATIONS

Below, we provide additional visualizations of normal and abnormal relationship graphs for visual features, text embeddings, and our EGO-fused representation on selected video samples.

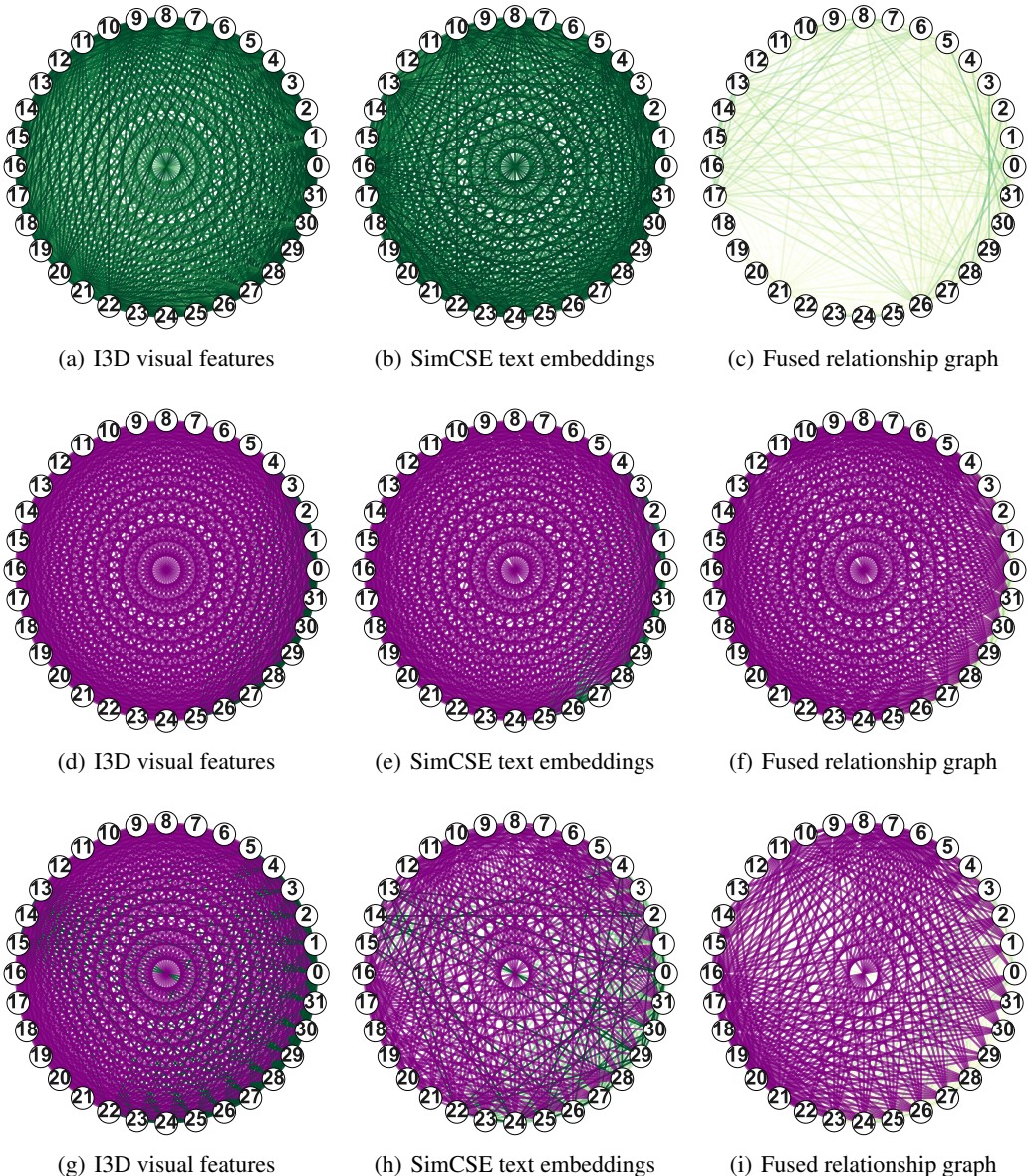

Figure 6: Comparison of relationship graphs on ShanghaiTech. The graphs are constructed using cosine similarity to represent relationships among features: (*first column*) visual features, (*second column*) text embeddings, and (*third column*) the fused graph, which integrates both modalities. In each graph, nodes represent clip-level (or unit-level) features, with numbers indicating the sequence order of the video clips. Edges, shown in green, represent cosine similarity between features, with darker shades indicating stronger connections. Anomaly nodes and their connections are highlighted in purple. The fused relationship graph, generated using our EGO fusion method, effectively integrates visual and textual information into a unified structure (see Figures (c), (f), and (i) for more details).

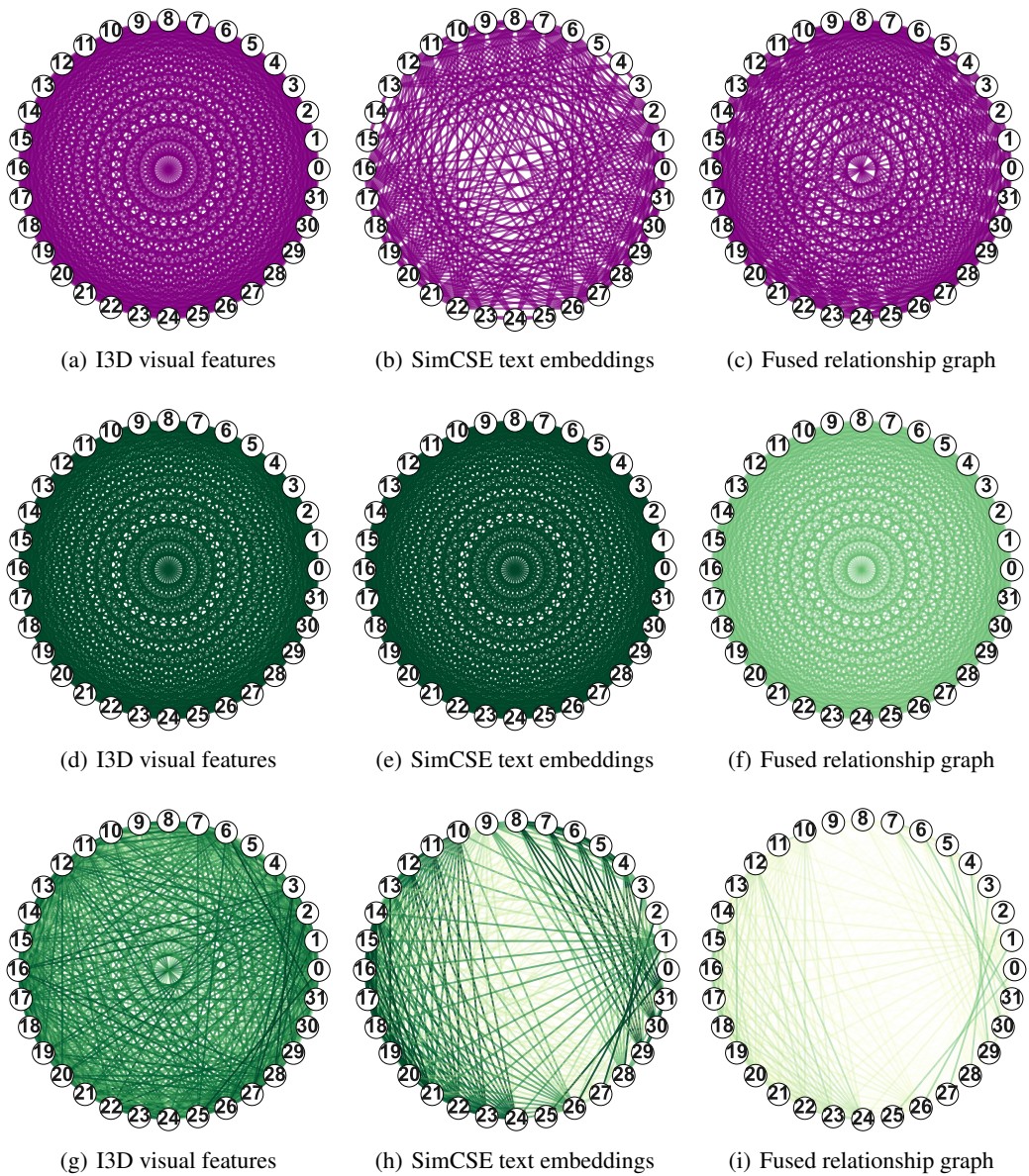

(a) I3D visual features      (b) SimCSE text embeddings      (c) Fused relationship graph

(d) I3D visual features      (e) SimCSE text embeddings      (f) Fused relationship graph

(g) I3D visual features      (h) SimCSE text embeddings      (i) Fused relationship graph

Figure 7: Comparison of relationship graphs: the first row shows UCSD Ped2, the second row shows CUHK Avenue, and the third row shows Street Scene. The graphs are constructed using cosine similarity to represent relationships among features: (*first column*) visual features, (*second column*) text embeddings, and (*third column*) the fused graph, which integrates both modalities. In each graph, nodes represent clip-level (or unit-level) features, with numbers indicating the sequence order of the video clips. Edges, shown in green, represent cosine similarity between features, with darker shades indicating stronger connections. Anomaly nodes and their connections are highlighted in purple. The fused relationship graph, generated using our EGO fusion method, effectively integrates visual and textual information into a unified structure (see Figures (c), (f), and (i) for more details).

