# OpenReview forum: "Learnable Expansion of Graph Operators for Multi-Modal Feature Fusion"
_ICLR.cc/2025/Conference — ICLR 2025 Poster_

### Official Review · Reviewer_Po6J · 2024-10-21

**Soundness:** 3
**Presentation:** 3
**Contribution:** 3
**Rating:** 6
**Confidence:** 3

**Summary:**

LEGO is a novel graph-based feature fusion framework integrating multi-representational, multi-modal, and multi-domain relationships through relationship graphs, enhancing feature representations. It introduces a learnable graph fusion operator that dynamically combines different graph powers, facilitating deeper feature interactions and balancing self- and inter-feature relationships. LEGO theoretically connects graph fusion with multilinear polynomials, providing valuable insights into feature interactions. Empirical evaluations in video anomaly detection demonstrate that LEGO outperforms traditional feature-level fusion methods.

**Strengths:**

1. The proposed framework seamlessly combines visual and text features, leveraging their complementary information to enhance anomaly detection. It utilizes graph power expansion and dynamically learns optimal weights for merging different graph powers, allowing the model to prioritize relevant relationships.
3. LEGO requires significantly fewer parameters than traditional methods like MTN fusion, resulting in faster training times.
4. LEGO consistently outperforms baseline and state-of-the-art methods across multiple datasets.

**Weaknesses:**

(See the questions section below)

**Questions:**

1. Given that graph power expansion and tensor operations can be computationally intensive, how does LEGO Fusion scale with larger datasets?
2. Since LEGO Fusion relies heavily on high-quality feature extraction, how does the framework perform when using features from models that may not be as robust or are trained on different datasets?
3. How does LEGO Fusion handle noisy or irrelevant features within the input data?

---

### Official Review · Reviewer_Fqv1 · 2024-10-27

**Soundness:** 2
**Presentation:** 3
**Contribution:** 3
**Rating:** 6
**Confidence:** 4

**Summary:**

This article focuses on multimodal feature fusion. The paper establishes separate graphs for each individual modality and then performs intra-graph convolution on each modality's graph, finally fusing the node features of different convolution depths. The authors believe that this approach can effectively achieve deeper fusion of multimodal features and avoid feature redundancy.

**Strengths:**

The illustration of the method is clear, and this method is easy to follow. The performance is credible.

**Weaknesses:**

Currently, there are many methods based on cross-modal graph neural networks that create their own graph networks for features from different modalities, and then use inter-graph convolution to obtain cross-modal embeddings, facilitating feature propagation and fusion in the process.

Compared to these methods, the improvements presented in this paper seem to consist only of weighted sums at different convolution depths. The authors need to explain the advantages of this approach over previous methods.

[1] Yu H, Yao F, Lu W, et al. Text-image matching for cross-modal remote sensing image retrieval via graph neural network[J]. IEEE Journal of Selected Topics in Applied Earth Observations and Remote Sensing, 2022, 16: 812-824.
[2] He Y, Liu X, Cheung Y M, et al. Cross-graph attention enhanced multi-modal correlation learning for fine-grained image-text retrieval[C]//Proceedings of the 44th international ACM SIGIR conference on research and development in information retrieval. 2021: 1865-1869.

**Questions:**

Please refer to weaknesses.

---

### Official Review · Reviewer_jZ3p · 2024-11-01

**Soundness:** 3
**Presentation:** 2
**Contribution:** 3
**Rating:** 6
**Confidence:** 4

**Summary:**

The paper introduces a graph-based method for multi-modal feature fusion in computer vision tasks. It shifts from high-dimensional feature spaces to interpretable graph spaces by constructing relationship graphs. This method leverages graph power expansions to capture multi-level interactions and introduces a learnable graph fusion operator for dynamic integration of graph powers. The approach is validated through experiments on video anomaly detection, demonstrating improved performance and deeper insights compared to traditional fusion techniques.

**Strengths:**

1. The graph-based fusion approach is innovative as it focuses on relationship-centric fusion, potentially capturing deeper interactions and structural relationships.
2. The use of graph power expansions to model multi-hop connections is a strong point.
3. Introducing a learnable weight matrix to dynamically integrate different graph powers is a key advancement.
4. This paper is easy to follow.

**Weaknesses:**

1. While the method is sophisticated, its complexity might hinder interpretability. The relationship between graph power expansions and their real-world applications could be elaborated further for clarity.
2. The scalability of the graph-based approach for large-scale datasets or real-time applications is not fully addressed. More discussion on computational efficiency would strengthen the paper.
3. The integration of the LEGO method with existing machine learning frameworks or pipelines is not clearly addressed. If the possible combination of LEGO and the existing method is given, it will help the method to be widely used.

**Questions:**

Please refer to the weakness.

---

### Meta-Review · Area_Chair_tmAU · 2024-12-22

**Metareview:**

This paper introduces LEGO, a novel feature fusion framework that enhances feature representations by integrating diverse relationships through relationship graphs. It introduces a dynamic graph fusion operator for deeper feature interactions, outperforming traditional methods in video anomaly detection. The reviewers expressed concerns about the novelty w.r.t. the previous methods, efficiency, results, as well as a few clarifications on the details. After the rebuttal, all reviewers lean towards the positive side by recommending borderline acceptance. The AC agrees with this recommendation and would like to recommend acceptance.

**Additional Comments On Reviewer Discussion:**

Before rebuttal, the paper receives recommendations of 6,5,6. After the rebuttal, all the reviewers acknowledged that they were satisfied with the rebuttal, leading to the increase of scores to 6, 6, 6. The AC found that the rebuttal indeed clearly addressed the reviewers' questions with sufficient evidence. The paper can be improved by incorporating the additional results and discussion provided during the rebuttal period.

---

### Decision · Program_Chairs · 2025-01-22

Accept (Poster)